# Do Metabolically Healthy People with Obesity Have a Lower Health-Related Quality of Life? A Prospective Cohort Study in Taiwan

**DOI:** 10.3390/jcm10215117

**Published:** 2021-10-30

**Authors:** Yi-Hsuan Lin, Hsiao-Ting Chang, Yen-Han Tseng, Harn-Shen Chen, Shu-Chiung Chiang, Tzeng-Ji Chen, Shinn-Jang Hwang

**Affiliations:** 1Department of Family Medicine, Cheng Hsin General Hospital, Taipei City 112401, Taiwan; yihsuan723@gmail.com; 2Faculty of Medicine, School of Medicine, National Yang Ming Chiao Tung University, Taipei City 112304, Taiwan; yenhan213@gmail.com (Y.-H.T.); chenhs@vghtpe.gov.tw (H.-S.C.); tjchen@vghtpe.gov.tw (T.-J.C.); sjhwang@vghtpe.gov.tw (S.-J.H.); 3Department of Family Medicine, Taipei Veterans General Hospital, Taipei City 11217, Taiwan; 4Department of Chest Medicine, Taipei Veterans General Hospital, Taipei City 11217, Taiwan; 5Division of Endocrinology and Metabolism, Department of Internal Medicine, Taipei Veterans General Hospital, Taipei City 11217, Taiwan; 6Institute of Hospital and Health Care Administration, School of Medicine, National Yang Ming Chiao Tung University, Taipei City 112304, Taiwan; scchiang0g@gmail.com

**Keywords:** obesity, metabolic health, metabolically healthy obesity, quality of life

## Abstract

The association between metabolically healthy obesity (MHO) and health-related quality of life (HRQOL) has not been thoroughly evaluated. This study enrolled 906 adult participants aged 35–55 years between 2009 and 2010 in Northern Taiwan; 427 participants were followed up after eight years. Normal weight, overweight, and obesity were evaluated via body mass index. Metabolic health was defined as the absence of cardiometabolic diseases and having ≤1 metabolic risk factor. HRQOL was evaluated using the 36-Item Short Form Health Survey (SF-36), Taiwan version. Generalized linear mixed-effects models were used to analyze the repeated, measured data with adjustment for important covariates. Compared with metabolically healthy normal weight individuals, participants with metabolically unhealthy normal weight and obesity had a significantly poorer physical component summary score (β (95% CI) = −2.17 (−3.38–−0.97) and −2.29 (−3.70–−0.87), respectively). There were no significant differences in physical and mental component summary scores among participants with metabolically healthy normal weight, overweight, and obesity. This study showed that metabolically healthy individuals with obesity and normal weight had similar HRQOL in physical and mental component summary scores. Maintaining metabolic health is an ongoing goal for people with obesity.

## 1. Introduction

Obesity is a major threat to public health worldwide. According to the World Health Organization, the global prevalence of obesity has nearly tripled since 1975, and there were more than 650 million adults with obesity in 2016 [1]. The global death and disability-adjusted life years (DALYs) attributable to high body mass index (BMI) have doubled since 1990 [2]. Obesity increases the risk of diabetes, cardiovascular diseases, dementia, and some cancers [3], and results in substantial medical costs in both developed and developing countries [4].

A subset of people with obesity without metabolic abnormalities have attracted attention since 2001 [5]. Metabolically healthy obesity (MHO) typically refers to obesity free from hypertension, hyperglycemia, and dyslipidemia [6]. Although studies about MHO have emerged in the last two decades, there are no universal diagnostic criteria for MHO. Obesity is usually defined by BMI, waist circumference, and total body fat percentage [6]. The components of metabolic health have been thought to include systolic or diastolic blood pressure, fasting glucose, and levels of serum triglyceride and high-density lipoprotein cholesterol (HDL-C) [6]. Other studies have adopted insulin resistance, waist circumference, white blood cell count, and C-reactive protein, among other factors [6]. In 2019, Smith et al. proposed an integrated set of criteria to define MHO, which included the absence of cardiometabolic diseases and healthy cardiometabolic profiles [7].

However, MHO does not always mean truly healthy; rather, it typically refers to having fewer metabolic syndrome components than healthy people [7]. The risk of cardiovascular disease and type 2 diabetes for individuals with MHO is lower than that of individuals with metabolically unhealthy obesity but higher than that of people with metabolically healthy normal weight [8,9]. MHO may represent a period of transition between metabolically healthy to unhealthy states. Favorable cardiovascular outcomes were observed in MHO subjects who could maintain metabolic health [10]. Despite much evidence on MHO and its health consequences, most studies have focused on medical outcomes. The effects of MHO on psychosocial aspects have not been thoroughly evaluated.

Health does not merely mean free from physical illness, mental and social wellbeing are also essential components. Health-related quality of life (HRQOL) represents a comprehensive evaluation of physical, mental, and social aspects of one’s wellbeing [11]. Many tools have been developed to evaluate HRQOL, such as the 36-Item Short Form Health Survey (SF-36), which is the instrument most widely used for this purpose [12]. A past meta-analysis showed different patterns of impaired physical and mental aspects of HRQOL in people with obesity [13]. However, research on MHO and HRQOL is rare. In 2017, Lopez-Garcia et al. reported that people with MHO scored lower in the physical component of HRQOL than those with healthy normal weight in Spain [14], although the study did not consider changes in metabolic health and BMI. Furthermore, although studies from Korea and Singapore reported that people with obesity had worse HRQOL than normal-weight people [15,16], Asian data for MHO and HRQOL is lacking. We aimed to explore the prospective association between MHO and HRQOL in the Taiwanese population and hoped to provide more evidence on this important topic.

## 2. Materials and Methods

### 2.1. Study Design and Participants

This study used a community-based, prospective cohort design. The study was approved by the Research Ethics Committee of Taipei Veteran General Hospital (Taipei City, Taiwan). During 2009–2010, residents aged 35–55 years and living for more than six months in the Shihpai area (Shilin and Beitou district) in Taipei, Taiwan, were candidates for study participation. Randomized sampling was conducted; subsequently, residents who agreed to participate signed an informed consent form before recruitment. At baseline (2009–2010), 906 participants were enrolled in this study. Basic demographic data, medical history, BMI, HRQOL, blood laboratory tests for metabolic health risk factors, including fasting glucose, triglyceride, and high-density lipoprotein cholesterol (HDL-C), and lifestyle habits, including smoking, alcohol consumption, and physical activity were evaluated at baseline. There were 427 participants who completed the follow-up evaluation between 2017 and 2018, including the repeated assessment of metabolic health, BMI, HRQOL, and other covariates. The mean follow-up period was 8.6 ± 0.6 years. However, participants with missing data at the follow-up assessment were still preserved in the analytical sample [17]. A total of 906 participants with complete data at baseline were included for the analyses.

### 2.2. Definition of Overweight/Obesity and Metabolic Health

Overweight and obesity were evaluated by measuring participants’ BMI (kg/m^2^). According to the Health Promotion Administration, Ministry of Health and Welfare, Taiwan [18], we defined normal weight as BMI < 24 kg/m^2^, overweight as 24 kg/m^2^ ≤ BMI < 27 kg/m^2^, and obesity as BMI ≥ 27 kg/m^2^.

The criteria of metabolic health were based on those in Smith’s 2019 publication [7], which were the absence of cardiometabolic diseases and healthy cardiometabolic profiles. We adopted the following modified criteria: (A) absence of hypertension, hyperlipidemia, type 2 diabetes, coronary artery diseases, cerebral vascular diseases, and peripheral vascular diseases; (B) presence of ≤1 of the following metabolic risk factors [19]: (a) hypertension: blood pressure ≥ 130/85 mmHg, or on drug treatment, (b) hyperglycemia: serum fasting glucose ≥ 100 mg/dL, or receiving drug treatment, (c) hypertriglyceridemia: serum triglyceride ≥ 150 mg/dL, or on drug treatment, and (d) low serum HDL-C: HDL-C < 40 mg/dL for men or <50 mg/dL for women, or on drug treatment.

The combination of metabolic health and BMI resulted in six categories: metabolically healthy normal weight (MHNW), metabolically healthy overweight (MHOW), metabolically healthy obesity (MHO), metabolically unhealthy normal weight (MUNW), metabolically unhealthy overweight (MUOW), and metabolically unhealthy obesity (MUO).

### 2.3. Evaluation of Health-Related Quality of Life

As it is known to have good validity and reliability, the Taiwan version of SF-36 was used to evaluate HRQOL in this study [20]. The SF-36 questionnaires are self-reported and can be calculated for eight domains: physical functioning, role participation with physical health problems (role-physical), bodily pain, general health, vitality, social functioning, role participation with emotional health problems (role-emotional), and mental health domains. The scores of each domain range from 0 to 100, while a higher score represents a better HRQOL. According to the scoring procedures of the user’s manual for the SF-36, the eight SF-36 domains can be converted to physical component summary (PCS) and mental component summary (MCS) scores by differential weighting and transformation [21]. Higher PCS and MCS scores indicate better physical and mental health status.

### 2.4. Covariates

Basic demographic data, medical history, marital status, levels of education, and health habits, including physical activity, cigarette smoking, and alcohol consumption were self-reported. Physical activity was evaluated with the International Physical Activity Questionnaire (IPAQ) Short-Form, Taiwan version [22]. Participants reported how much time they spent walking or performing moderate and vigorous physical activity in the past week. Physical activity can be quantified into metabolic equivalent units (MET-minutes/week) and categorized into high, moderate, and low physical activity groups according to the scoring principles of the IPAQ [23]. Cigarette smoking was grouped into nonsmokers, smokers, and ex-smokers (individuals who had quit cigarette smoking). Alcohol consumption was measured by asking participants if they drank alcohol.

Blood pressure (mmHg), height (cm), weight (kg), and waist circumference (cm) were measured by well-trained nurses. We checked blood pressure in a seated position three times to obtain an average blood pressure for analyses. Blood tests, including fasting glucose (mg/dL), serum HDL-C (mg/dL), and triglycerides (mg/dL), were collected from each participant under a fasting status longer than eight hours. Laboratory tests were performed by automated chemistry analyzers at a central laboratory at Taipei Veterans General Hospital.

### 2.5. Statistical Analysis

For basic descriptive analyses, we used chi-squared tests and unbalanced ANOVA tests to compare the categorical and continuous variables, respectively, among metabolic health/BMI groups. The repeated measured data were autocorrelated for the same participant. Generalized linear mixed-effects models (GLMMs) and R-side analysis with a random intercept at the individual level [24], were used for the main analyses to estimate the β coefficient and the 95% confidence interval (95% CI) for the association between metabolic health/BMI groups and HRQOL. We adjusted age (at baseline), sex, marital status, level of education, cigarette smoking, alcohol consumption, physical activity, and follow-up years. For variables that showed significant differences among metabolic health/BMI groups at baseline, subgroup analyses were performed (stratified by sex, marital status, smoking, and alcohol consumption).

We further conducted sensitivity analyses by adopting a different definition of metabolic health. In addition to the criteria of metabolic health in the main analyses, abdominal obesity was included in the definition of metabolic health in the sensitivity analyses. Abdominal obesity was defined as waist circumference ≥90 cm in men and ≥80 cm in women for Asian populations [19]. Participants were deemed as metabolically healthy if they had no known cardiometabolic diseases and had ≤ two metabolic risk factors. All statistical analyses were performed by using SAS version 9.4 (SAS Institute Inc., Cary, NC, USA). A two-tailed *p*-value of <0.05 was considered statistically significant.

## 3. Results

Table 1 compares the demographic characteristics among different metabolic health/BMI groups at baseline. The mean age of study participants at baseline was 46.9 ± 5.5 years. Overall, 559 (61.7%) of the participants were women. The prevalence of MHO in all study participants was 6.5% at baseline and 4.9% at eight-year follow-up, respectively. The proportion of MHO in participants with obesity was 38.1% at baseline and 26.3% after eight years. Compared with metabolically unhealthy participants, metabolically healthy adults were younger (mean age 45.9–46.9 vs. 48.0–48.8 years), more likely to be women (ratio of women 52.9–78.0% vs. 34.4–61.9%), and less likely to be smokers (ratio of smokers 11.0–13.6% vs. 13.6–26.0%). Participants with MHNW had the lowest ratio of being married (79.8% vs. 84.2–91.4%). There were no significant differences in physical activity and levels of education among metabolic health/BMI groups. At baseline, participants with MHO had higher scores of PCS, role-physical, bodily pain, general health, and social functioning. The demographic characteristics at follow-up of 427 participants with complete follow-up assessment are shown in Appendix A. After the nine-year follow-up, the scores of vitality, social functioning, mental health, and MCS improved (mean score at baseline and follow-up: 47.8 to 54.0, 31.6 to 50.3, 38.5 to 47.0, and 38.0 to 48.7, respectively; Table 1 and Appendix A). There were no significant differences in PCS, MCS, and the eight domains of HRQOL among different metabolic health/BMI groups at the follow-up assessment.

Compared with MHNW individuals, participants with MUNW and MUO had a lower PCS score (β (95% CI) = −2.17 (−3.38, −0.97) and −2.29 (−3.70, −0.87), Table 2). Participants with MHOW and MHO had similar PCS scores to MHNW individuals. For the mental aspect of HRQOL, the status of metabolic health/BMI groups did not significantly affect MCS scores.

Figure 1 shows the association between metabolic health/BMI groups and the eight SF-36 domains. Compared with MHNW individuals, MHOW participants had lower physical functioning and role-physical scores (β (95% CI) = −1.10 (−1.87, −0.34) and −1.55 (−3.09, −0.02), respectively). Metabolically unhealthy adults (no matter the BMI grouping) also had worse performance on physical functioning scores than MHNW individuals (β (95% CI) = −0.99 (−1.85, −0.12), −1.01 (−1.90, −0.11) and −1.78 (−2.81, −0.74) for MUNW, MUOW, and MUO, respectively). However, compared with MHNW individuals, higher scores of bodily pain and general health were observed in participants with MHO (β (95% CI) = 4.25 (1.46, 7.03) and 3.47 (0.30, 6.64), respectively). The comparisons of HRQOL between individuals with metabolically healthy overweight and metabolically unhealthy overweight, as well as metabolically healthy obesity and metabolically unhealthy obesity are shown in Appendix A.

In the subgroup analyses, MHOW participants had a lower PCS score among nonsmokers and ex-smokers (β (95% CI) = −1.29 (−2.53, −0.05), Table 3). The worse performance of MUNW participants on PCS score was stronger for smokers than nonsmokers and ex-smokers (β (95% CI) = −4.82 (−8.12, −1.52) and −1.90 (−3.16, −0.64), respectively).

In the sensitivity analyses, the prevalence of MHO in study participants was 6.7% at baseline and 5.6% at eight-year follow-up. The proportion of MHO in participants with obesity was 39.4% at baseline and 30% after eight years. In general, the sensitivity analyses showed similar results to those of the main analyses. There were no significant differences in PCS and MCS scores among MHNW, MHOW, and MHO groups (Table 4). However, in addition to MHOW adults, participants with MHO also had a lower physical functioning score. The positive scores of bodily pain and general health of MHO participants were still significant compared to MHNW people in the sensitivity analyses.

## 4. Discussion

Our study found one person with MHO in every 2.6 individuals with obesity at baseline. Individuals with MHOW and MHO had similar PCS and MCS scores, compared with MHNW adults. Poor PCS scores were associated with MUNW and MUO status. Participants with MHOW had poor physical functioning domain scores, while adults with MHO performed better on bodily pain and general health, compared with MHNW individuals.

The reported prevalence of MHO varied greatly in prior studies because they implemented different criteria for MHO. Rey-López et al. reviewed studies on MHO from before 2013 and reported the overall prevalence of MHO in the population with obesity, which ranged from 6–75% [6]. Our results showed the prevalence of MHO in participants with obesity was 38.1%, which was very close to Kuk and Ardern’s 2009 report [25]. Kuk and Ardern analyzed 6,011 American adults aged 18–65 years from the Third National Health and Nutrition Examination Survey by using ≤1 metabolic syndrome factor. as in our study; they reported an MHO prevalence of 38.4%.

Our results showed that metabolically healthy participants, regardless of BMI status, had similar HRQOL in PCS and MCS. Lopez-Garcia et al. found similar MCS scores among all metabolic health/BMI groups [14], which were comparable with our results. However, Lopez-Garcia et al. also reported that individuals with MHO had poorer PCS performance, compared with MHNW adults. In Lopez-Garcia’s report, metabolic health and BMI were evaluated at baseline. However, metabolic health and BMI status may change over time. In our study, the prevalence of MHO decreased during the study period. Meanwhile, the proportion of MUO increased. Poorer outcomes in the baseline MHO group would be observed more easily if we ignored the MHO individuals who may change to MUO over time [8]. In our study, we evaluated the MHO status at baseline and at an eight-year follow-up; the GLMMs contained the changes of metabolic health status and the effects of repeated measures in the analytical models. We did not find worse HRQOL in PCS and MCS among individuals with MHO, compared with those with MHNW.

MHO is usually a transient but not permanent state. The North West Adelaide Health Study followed 454 adults with MHO for a median follow-up of 8.2 years [10]. About one-third of individuals with MHO did not maintain metabolically healthy status during the follow-up period, and these unstable adults with MHO were found to be at an increased risk of diabetes and metabolic disorders. Other cohort studies also found that baseline MHO is transient [26,27]. When MHO changed to MUO, the risk of cardiovascular events also increased. Conversely, for adults with MHO throughout, the risk of cardiovascular diseases was similar to those with MHNW throughout. These results were consistent with our findings. Past studies found the fat distribution differed between people with MHO and MUO [28,29], which might explain why metabolic health had a greater influence on health outcomes than BMI.

Interestingly, we found that adults with MHO had better scores on bodily pain and general health, compared with MHNW individuals. Although previous studies found a positive association between obesity and chronic pain via multifactorial linkages such as mechanical stresses and inflammatory state [30], another study showed different results. Price et al. reported that people with obesity were less sensitive to pain in the area with excessive subcutaneous fat (abdomen), compared with individuals without obesity [31], which may be due to decreased nerve fiber density and increased anti-inflammatory cytokines in adipose tissue [31,32]. However, Price et al. also observed the sensitivity of pain on the forehead and hand was not altered in obese participants. In our study, the questionnaire of SF-36 evaluated the general perception of pain without the differentiation for the location of pain. Further studies are needed to investigate the perception of pain among people of MHO with considerations of the kinds and locations of pain. Previous studies demonstrated a relationship between obesity and poorer self-rated health [33,34]. Nevertheless, metabolic health has not been adjusted in past studies. More research is warranted to clarify the association between MHO and general and self-rated health.

This study has several strengths. First, research concerning MHO and HRQOL is rare; moreover, our study is the first one examining an Asian population. We enrolled participants from communities with a long follow-up period. Second, we used a widely validated tool, the SF-36, to evaluate HRQOL so that our results could be easily compared with other research. Third, the evaluations of metabolic health, BMI, HRQOL, and lifestyle covariates were measured repeatedly. We used GLMMs for longitudinal data and adjusted for important covariates. Fourth, we conducted sensitivity analyses by a different definition of metabolic health and our results were consistent.

There were some limitations to this study. First, insulin resistance data were not available, so we could not use insulin resistance to define metabolic health. Second, the pattern of changes in metabolic health and BMI was not used as an independent variable because too many subgroups resulted in a small sample size in each subgroup in our study. We analyzed metabolic health and BMI status at baseline and at an eight-year follow-up in the analytical models (GLMMs) to consider dynamic changes in metabolic health and BMI. Third, the number of study participants in the subgroups of metabolic health/BMI was small. In addition, the study participants were recruited from the Shihpai area in Taipei, Taiwan so the generalizability of our findings might be limited. Further research with a larger representative cohort is needed.

## 5. Conclusions

Our study found that individuals with MHO had similar HRQOL in both PCS and MCS, compared with people with MHNW. Maintaining metabolic health to preserve HRQOL is important for individuals with MHO. Future research is warranted to clarify the association between the pattern of changes in MHO and HRQOL. More study is also required to investigate how to keep the status of metabolic health for people with MHO.

## Figures and Tables

**Figure 1 jcm-10-05117-f001:**
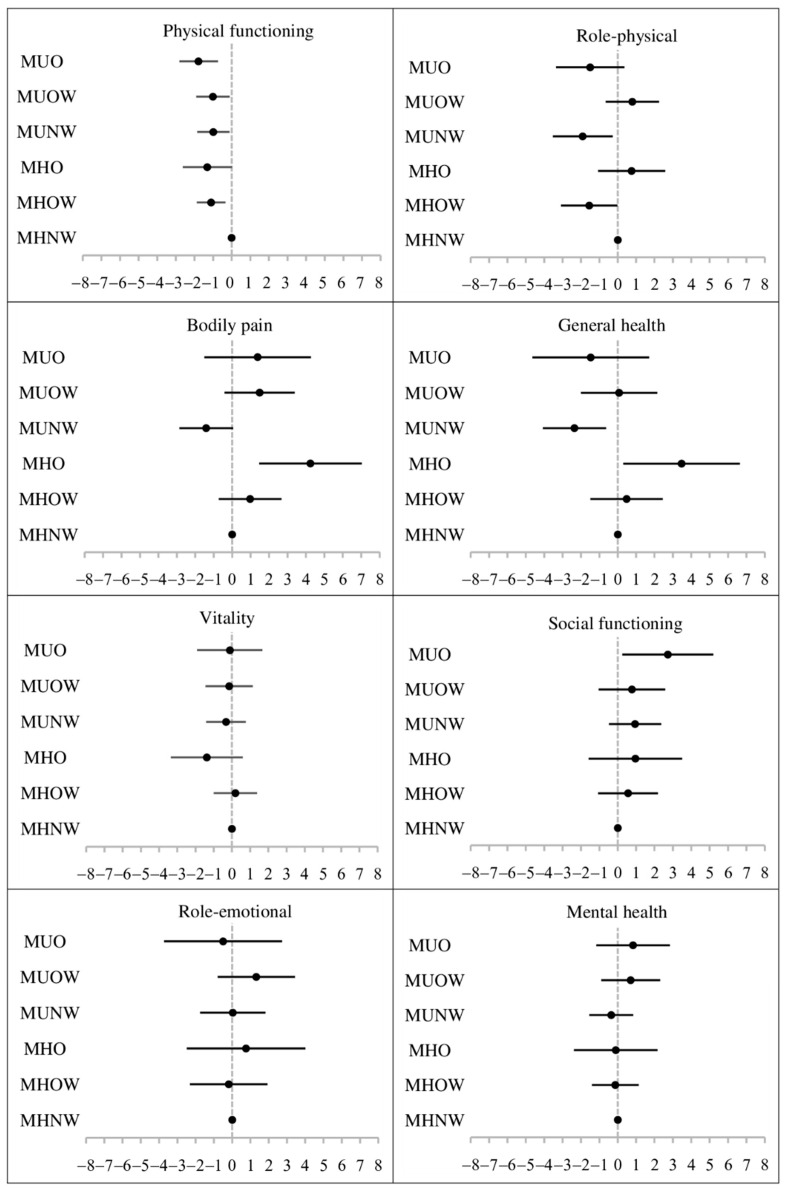
The association between metabolic health/BMI groups and eight domains of SF-36 (β coefficient and 95% confidence interval). Note. BMI = body mass index; MHNW = metabolically healthy normal weight (reference group); MHOW = metabolically healthy overweight; MHO = metabolically healthy obesity; MUNW = metabolically unhealthy normal weight; MUOW = metabolically unhealthy overweight; MUO = metabolically unhealthy obesity; SF-36 = the 36-Item Short Form Health Survey. The analyses were conducted by generalized linear mixed-effects models with adjustment for age, sex, marital status, level of education, smoking, alcohol consumption, groups of physical activity, and follow-up years.

**Table 1 jcm-10-05117-t001:** Demographic characteristics of participants at baseline, classified by the metabolic health/BMI groups at baseline.

	Total	Metabolically Healthy Normal Weight ^a^	Metabolically Healthy Overweight ^a^	Metabolically Healthy Obesity ^a^	Metabolically Unhealthy Normal Weight	Metabolically Unhealthy Overweight	Metabolically Unhealthy Obesity	*p*
	*n* = 906	*n* = 404 (44.6%)	*n* = 136 (15.0%)	*n* = 59 (6.5%)	*n* = 118 (13.0%)	*n* = 93 (10.3%)	*n* = 96 (10.6%)	
Age years, *n* (%)								
35–39	132 (14.6)	76 (18.8)	25 (18.4)	9 (15.3)	5 (4.2)	9 (9.7)	8 (8.3)	<0.001
40–49	457 (50.4)	218 (54.0)	67 (49.3)	30 (50.9)	55 (46.6)	41 (44.1)	46 (47.9)	
50–55	317 (35.0)	110 (27.2)	44 (32.4)	20 (33.9)	58 (49.2)	43 (46.2)	42 (43.8)	
Mean age (SD)	46.9 (5.5)	45.9 (5.6)	46.6 (5.6)	46.9 (5.5)	48.8 (4.7)	48.3 (5.5)	48.0 (5.3)	<0.001
Women, *n* (%)	559 (61.7)	315 (78.0)	72 (52.9)	33 (55.9)	73 (61.9)	33 (35.5)	33 (34.4)	<0.001
Mean BMI (SD), kg/m^2^	23.8 (3.5)	21.3 (1.7)	25.3 (0.9)	29.4 (2.5)	22.2 (1.5)	25.4 (0.8)	29.7 (2.7)	<0.001
Married, *n* (%)	764 (84.9)	320 (79.8)	124 (91.2)	53 (89.8)	102 (87.9)	85 (91.4)	80 (84.2)	0.004
Alcohol consumption, *n* (%)	139 (15.4)	52 (12.9)	24 (17.7)	9 (15.3)	15 (12.7)	25 (26.9)	14 (14.6)	0.03
Cigarette smoking, *n* (%)								
Nonsmokers	723 (80.0)	342 (85.1)	107 (79.3)	46 (78.0)	99 (83.9)	69 (74.2)	60 (62.5)	<0.001
Smokers	125 (13.8)	44 (11.0)	18 (13.3)	8 (13.6)	16 (13.6)	14 (15.1)	25 (26.0)	
Ex-smokers	55 (6.1)	16 (4.0)	10 (7.4)	5 (8.5)	3 (2.5)	10 (10.8)	11 (11.5)	
Physical activity ^b^, *n* (%)								
Low	318 (35.1)	146 (36.1)	49 (36.0)	18 (30.5)	40 (33.9)	28 (30.1)	37 (38.5)	0.88
Moderate	403 (44.5)	173 (42.8)	62 (45.6)	28 (47.5)	50 (42.4)	49 (52.7)	41 (42.7)	
High	185 (20.4)	85 (21.0)	25 (18.4)	13 (22.0)	28 (23.7)	16 (17.2)	18 (18.8)	
Education, *n* (%)								
Illiterate/ elementary school	16 (1.8)	7 (1.8)	3 (2.2)	2 (3.4)	0	1 (1.1)	3 (3.1)	0.12
Senior/junior high school	324 (35.9)	125 (31.2)	54 (39.7)	25 (42.4)	43 (36.4)	33 (35.5)	44 (45.8)	
University and above	563 (62.4)	269 (67.1)	79 (58.1)	32 (54.2)	75 (63.6)	59 (63.4)	49 (51.0)	
SF-36 (mean ± SD)								
PCS	54.7 (6.9)	55.4 (6.4)	54.3 (7.5)	56.8 (6.4)	52.3 (7.7)	55.5 (5.4)	53.6 (7.5)	<0.001
MCS	38.0 (5.0)	37.5 (5.5)	38.3 (4.6)	37.8 (3.9)	38.3 (4.8)	38.5 (4.2)	38.6 (4.8)	0.17
Physical functioning	53.1 (5.1)	53.6 (4.8)	52.6 (4.9)	52.7 (5.9)	51.9 (5.6)	53.4 (4.7)	52.8 (5.3)	0.01
Role-physical	51.8 (8.4)	52.2 (8.1)	50.6 (9.4)	54.1 (6.0)	50.0 (9.7)	53.2 (7.0)	51.2 (8.3)	0.007
Bodily pain	52.9 (8.2)	53.1 (7.7)	53.1 (8.6)	55.0 (7.9)	50.5 (8.8)	54.3 (7.5)	52.0 (9.6)	0.002
General health	47.3 (9.2)	47.6 (8.8)	47.7 (9.9)	51.3 (7.9)	44.7 (10.1)	47.6 (7.7)	45.6 (9.8)	<0.001
Vitality	47.8 (4.6)	47.9 (4.6)	48.4 (4.7)	46.7 (4.4)	47.1 (4.0)	47.9 (4.8)	47.9 (4.7)	0.13
Social functioning	31.6 (4.2)	31.3 (4.0)	31.8 (3.6)	32.8 (4.2)	31.4 (4.4)	31.5 (4.1)	32.5 (5.2)	0.04
Role-emotional	50.4 (9.8)	49.9 (10.1)	50.0 (10.3)	52.1 (7.1)	50.5 (10.2)	51.7 (8.0)	50.4 (10.4)	0.44
Mental health	38.5 (4.5)	38.5 (4.7)	38.7 (4.0)	37.6 (3.5)	37.9 (4.7)	39.3 (4.6)	38.7 (4.0)	0.14

Note. SD = standard deviation; BMI = body mass index; SF-36 = the 36-Item Short Form Health Survey; PCS = physical component summary; MCS = mental component summary. Chi-squared tests and Fisher’s exact tests were used for categorical variables. Unbalanced ANOVA tests were used for continuous variables. ^a^ Metabolic health was defined as (1) absence of known chronic diseases including hypertension, hyperlipidemia, diabetes, coronary artery disease, stroke, and vascular diseases; (2) presence of ≤1 metabolic risk factor, including hypertension, hyperglycemia, hypertriglyceridemia, and low serum high-density lipoprotein cholesterol. ^b^ Physical activity was evaluated by the International Physical Activity Questionnaire (IPAQ) Short-Form, Taiwan version.

**Table 2 jcm-10-05117-t002:** The association between metabolic health/BMI groups and health-related quality of life.

	Physical Component Summary		Mental Component Summary	
	β (95% CI)	*p*	β (95% CI)	*p*
Status of metabolic health ^a^ and BMI				
Metabolically healthy normal weight	Reference group		Reference group	
Metabolically healthy overweight	−0.91 (−2.01, 0.20)	0.13	0.16 (−1.01, 1.34)	0.78
Metabolically healthy obesity	1.09 (−0.45, 2.62)	0.17	−0.56 (−2.07, 0.96)	0.47
Metabolically unhealthy normal weight	−2.17 (−3.38, −0.97)	<0.001	0.50 (−0.73, 1.73)	0.43
Metabolically unhealthy overweight	−0.53 (−1.62, 0.55)	0.34	0.81 (−0.58, 2.20)	0.26
Metabolically unhealthy obesity	−2.29 (−3.70, −0.87)	0.002	0.97 (−0.36, 2.30)	0.16
Age	−0.08 (−0.15, −0.01)	0.03	0.06 (−0.02, 0.13)	0.13
Women	−1.44 (−2.39, −0.50)	0.003	−0.28 (−1.22, 0.66)	0.56
Marital status				
Single/Divorced/Separated/Widowed/Others	Reference group		Reference group	
Married	−0.04 (−1.21, 1.13)	0.95	0.76 (−0.36, 1.89)	0.19
Education				
Illiterate/elementary school	Reference group		Reference group	
Senior/junior high school	3.44 (0.70, 6.19)	0.01	−1.25 (−3.86, 1.37)	0.35
University and above	4.46 (1.72, 7.19)	0.001	−0.83 (−3.45, 1.79)	0.54
Cigarette smoking				
Nonsmokers	Reference group		Reference group	
Smokers	−1.10 (−2.39, 0.19)	0.10	−2.06 (−3.37, −0.76)	0.004
Ex-smokers	1.87 (0.63, 3.10)	0.005	−0.48 (−2.14, 1.17)	0.57
Alcohol	1.24 (0.25, 2.24)	0.02	0.36 (−0.85, 1.57)	0.56
Physical activity ^b^, *n* (%)				
Low	Reference group		Reference group	
Moderate	0.81 (0.04, 1.59)	0.04	−1.17 (−2.05, −0.28)	0.01
High	2.27 (1.28, 3.27)	<0.001	−1.39 (−2.53, −0.26)	0.02
Follow-up year	−0.08 (−0.17, 0.01)	0.09	0.56 (0.48, 0.64)	<0.001

^a^ Metabolic health was defined as (1) absence of cardiometabolic diseases; (2) presence of ≤ 1 metabolic risk factor, including hypertension, hyperglycemia, hypertriglyceridemia, and low serum high-density lipoprotein cholesterol. ^b^ Physical activity was evaluated by the International Physical Activity Questionnaire (IPAQ) Short-Form, Taiwan version. Note. BMI = body mass index. The analyses were conducted by generalized linear mixed-effects models.

**Table 3 jcm-10-05117-t003:** The association between metabolic health/BMI groups and health-related quality of life, stratified by sex, marital status, smoking, and alcohol consumption at baseline.

	Metabolically Healthy Normal Weight	Metabolically Healthy Overweight	Metabolically Healthy Obesity	Metabolically Unhealthy Normal Weight	Metabolically Unhealthy Overweight	Metabolically Unhealthy Obesity	*p* _interactions_
Physical Component Summary							
Sex							
Men (*n* = 347)	Reference group	−0.36 (−2.05, 1.32)	0.13 (−2.31, 2.57)	−2.38 (−4.48, −0.28)	−0.48 (−2.03, 1.06)	−1.70 (−3.56, 0.16)	0.56
Women (*n* = 559)	Reference group	−1.09 (−2.67, 0.49)	2.14 (0.21, 4.08)	−2.04 (−3.52, −0.57)	−0.57 (−2.12, 0.98)	−3.14 (−5.45, −0.83)	
Marital status							
Married (*n* = 764)	Reference group	−0.72 (−1.94, 0.50)	1.15 (−0.49, 2.79)	−2.06 (−3.37, −0.75)	−0.24 (−1.38, 0.90)	−1.53 (−2.90, −0.15)	0.60
Single/Divorced/ Separated/ Widowed/Others (*n* = 136)	Reference group	0.92 (−2.29, 4.14)	3.02 (−1.40, 7.44)	−1.63 (−4.49, 1.24)	0.56 (−2.56, 3.68)	−3.27 (−8.21, 1.66)	
Smoking status							
Smokers (*n* = 125)	Reference group	1.94 (−1.25, 5.13)	−1.02 (−6.63, 4.59)	−4.82 (−8.12, −1.52)	−0.13 (−3.07, 2.81)	−1.53 (−4.25, 1.18)	0.001
Nonsmokers + Ex-smokers (*n* = 778)	Reference group	−1.29 (−2.53, −0.05)	1.37 (−0.17, 2.92)	−1.90 (−3.16, −0.64)	−0.57 (−1.73, 0.58)	−2.36 (−4.03, −0.69)	
Alcohol consumption							
Yes (*n* = 765)	Reference group	1.10 (−1.33, 3.53)	−0.50 (−4.16, 3.15)	1.08 (−1.38, 3.54)	−0.04 (−2.46, 2.38)	−1.96 (−5.10, 1.19)	0.04
No (*n* = 139)	Reference group	−0.78 (−2.05, 0.49)	1.87 (0.23, 3.50)	−2.31 (−3.63, −0.99)	0.17 (−1.04, 1.37)	−1.64 (−3.19, −0.09)	
Mental Component Summary							
Sex							
Men (*n* = 347)	Reference group	0.23 (−1.47, 1.93)	−1.16 (−3.61, 1.28)	0.63 (−1.53, 2.79)	−0.15 (−2.05, 1.76)	−0.08 (−1.95, 1.78)	0.23
Women (*n* = 559)	Reference group	−0.37 (−1.98, 1.24)	−0.47 (−2.48, 1.54)	0.36 (−1.13, 1.86)	1.80 (−0.31, 3.92)	2.39 (0.51, 4.26)	
Marital status							
Married (*n* = 764)	Reference group	0.18 (−1.00, 1.37)	−0.43 (−2.08, 1.21)	0.58 (−0.74, 1.91)	0.74 (−0.72, 2.20)	1.68 (0.37, 3.00)	0.22
Single/Divorced/ Separated/ Widowed/Others (*n* = 136)	Reference group	1.74 (−2.49, 5.97)	−1.41 (−5.64, 2.83)	1.02 (−2.21, 4.25)	3.43 (0.20, 6.65)	−2.48 (−5.82, 0.86)	
Smoking status							
Smokers (*n* = 125)	Reference group	−0.34 (−3.37, 2.70)	1.821 (−1.89, 5.54)	0.24 (−4.48, 4.95)	3.66 (−0.10, 7.43)	0.90 (−2.09, 3.88)	0.34
Nonsmokers + Ex-smokers (*n* = 778)	Reference group	0.22 (−1.04, 1.49)	−0.83 (−2.48, 0.82)	0.50 (−0.76, 1.77)	0.43 (−1.03, 1.90)	1.05 (−0.37, 2.47)	
Alcohol consumption							
Yes (*n* = 765)	Reference group	1.74 (−0.95, 4.43)	2.30 (−1.61, 6.22)	3.19 (−0.33, 6.72)	1.28 (−2.20, 4.75)	0.48 (−3.55, 4.51)	0.22
No (*n* = 139)	Reference group	−0.28 (−1.54, 0.97)	−1.01 (−2.52, 0.51)	−0.09 (−1.37, 1.19)	0.88 (−0.61, 2.37)	0.96 (−0.40, 2.31)	

Note. BMI = body mass index; The analyses were conducted by generalized linear mixed-effects models with adjustment for age, sex, marital status, level of education, smoking, alcohol consumption, groups of physical activity, and follow-up years.

**Table 4 jcm-10-05117-t004:** Sensitivity analyses to examine the association between metabolic health/BMI groups and health-related quality of life by using another definition of metabolic health.

	Metabolically Healthy ^a^ Normal Weight	Metabolically Healthy ^a^ Overweight	Metabolically Healthy ^a^ Obesity	Metabolically Unhealthy Normal Weight	Metabolically Unhealthy Overweight	Metabolically Unhealthy Obesity
PCS	Reference group	−0.79 (−1.90, 0.32)	1.09 (−0.42, 2.59)	−2.45 (−3.84, −1.06)	−0.21 (−1.33, 0.91)	−2.16 (−3.58, −0.74)
MCS	Reference group	0.53 (−0.60, 1.67)	−0.51 (−1.97, 0.94)	−0.30 (−1.50, 0.90)	−0.29 (−1.73, 1.15)	0.71 (−0.63, 2.04)
Physical functioning	Reference group	−0.98 (−1.71, −0.25)	−1.28 (−2.55, −0.01)	−1.27 (−2.30, −0.24)	−1.10 (−2.07, −0.14)	−1.74 (−2.78, −0.69)
Role-physical	Reference group	−1.02 (−2.48, 0.43)	0.80 (−1.02, 2.61)	−1.61 (−3.44, 0.22)	0.97 (−0.61, 2.55)	−1.29 (−3.12, 0.55)
Bodily pain	Reference group	0.91 (−0.74, 2.57)	4.09 (1.32, 6.85)	−2.39 (−4.02, −0.76)	1.52 (−0.40, 3.45)	1.19 (−1.68, 4.06)
General health	Reference group	0.63 (−1.21, 2.48)	3.39 (0.30, 6.48)	−2.90 (−4.87, −0.94)	0.12 (−2.04, 2.28)	−1.30 (−4.43, 1.82)
Vitality	Reference group	0.37 (−0.75, 1.49)	−1.38 (−3.27, 0.51)	−0.97 (−2.18, 0.23)	−0.98 (−2.32, 0.36)	−0.37 (−2.17, 1.43)
Social functioning	Reference group	0.38 (−1.19, 1.95)	0.62 (−1.87, 3.10)	−0.93 (−2.57, 0.70)	−0.63 (−2.51, 1.25)	1.65 (−0.78, 4.08)
Role-emotional	Reference group	0.34 (−1.68, 2.35)	0.82 (−2.51, 4.15)	0.62 (−1.25, 2.50)	1.02 (−1.28, 3.31)	−0.34 (−3.50, 2.83)
Mental health	Reference group	0.14 (−1.10, 1.38)	−0.34 (−2.53, 1.85)	−1.47 (−2.70, −0.24)	−0.33 (−2.02, 1.36)	0.47 (−1.54, 2.49)

^a^ Metabolic health was defined as (1) absence of cardiometabolic diseases; (2) presence of ≤2 metabolic risk factors, including hypertension, hyperglycemia, hypertriglyceridemia, low serum high-density lipoprotein cholesterol, and abdominal obesity. Note. BMI = body mass index; PCS = physical component summary; MCS = mental component summary. The analyses were conducted by generalized linear mixed-effects models with adjustment for age, sex, marital status, level of education, smoking, alcohol consumption, groups of physical activity, and follow-up years.

## Data Availability

The data used in this study are available on request from the corresponding author.

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
