# Peer review of "Do Metabolically Healthy People with Obesity Have a Lower Health-Related Quality of Life? A Prospective Cohort Study in Taiwan"

_jcm, 2021, doi:10.3390/jcm10215117_

Round 1

Reviewer 1 Report

Lin and colleagues determine prevalence of metabolically healthy obesity in 906 adults from Taiwan and whether health-related quality of life differs across groups separated by BMI and metabolically health cut-offs. Prevalence of metabolically healthy obesity was also assessed in 427 of the initial 906 participants after a 8.6 year follow-up period. At baseline, people in the metabolically healthy obesity group had a similar physical component summary score compared to metabolically healthy normal weight individuals. In contrast, participants with metabolically unhealthy normal weight and obesity had a significantly poorer physical component summary score compared to the metabolically healthy normal weight group. 

Comments:
1)    The rationale for inclusion of follow-up data is unclear. Was SF-36 completed at follow-up? If so please include this data. If not then the rational for presenting follow-up data should be more clearly defined. For example, was this to solely track changes in the prevalence of metabolically healthy obesity over time? 
2)    A Table with subject characteristics at follow-up would be helpful. 
3)    Comparing data to the MHNW group as the primary analysis makes sense but a secondary analysis comparing MHO and MUO would be helpful to determine whether outcomes differ between those groups. 
4)    In methods age range is 35-55 years but 30 to 60 yr data is presented in Table 1.
5)    In the introduction it is stated that “Metabolically healthy obesity (MHO) refers to obesity free from hypertension, hyperglycemia, and dyslipidemia.” As mentioned by the authors there is no universally used criteria for MHO. Please add “typically” before “refers to” or some similar wording and provide a reference at the end of this sentence.
6)    In the introduction it is mentioned that “MHO does not mean truly healthy; rather, it refers to having fewer metabolic syndrome components than healthy people.” Again this statement requires some nuance with “does not always mean” instead of “does not mean” and “typically” before “refers to” or similar wording added.

Author Response

On behalf of all the authors, we thank you very much for reviewing our manuscript. During the major revision, we found some mistakes in SAS code in the main analyses for domains of SF-36. We have corrected the errors. Fortunately, the significance of the statistical analyses and results are the same and only the effect size is altered. We revised Figure 1 and Table 4 accordingly. Thank you again for your valuable comments, which really helped us to improve this article. Please find below our point-by-point responses to the comments from the reviewers. Revisions were made by using track changes in the revised manuscript.

Response to reviewers’ comments:

Reviewer #1:

Comments and Suggestions for Authors

Lin and colleagues determine prevalence of metabolically healthy obesity in 906 adults from Taiwan and whether health-related quality of life differs across groups separated by BMI and metabolically health cut-offs. Prevalence of metabolically healthy obesity was also assessed in 427 of the initial 906 participants after a 8.6 year follow-up period. At baseline, people in the metabolically healthy obesity group had a similar physical component summary score compared to metabolically healthy normal weight individuals. In contrast, participants with metabolically unhealthy normal weight and obesity had a significantly poorer physical component summary score compared to the metabolically healthy normal weight group. 

Comments:
1)    The rationale for inclusion of follow-up data is unclear. Was SF-36 completed at follow-up? If so please include this data. If not then the rational for presenting follow-up data should be more clearly defined. For example, was this to solely track changes in the prevalence of metabolically healthy obesity over time? 

Response: Thanks for the reviewer’s comment. There were 906 participants enrolled at baseline, of them 427 participants completed the follow-up assessment after a mean follow-up of 8.6 ± 0.6 years. For the 427 participants, their metabolic health, BMI, HRQOL, and other covariates were assessed at follow-up. We used the generalized linear mixed-effects models (GLMMs) for analyses, which allowed for unbalanced data by incorporating all available information (Reference 17). We have added more descriptions in the 2.1. Study Design and Participants (page 2, 4th paragraph) as follows:

“At baseline (2009-2010), 906 participants were enrolled in this study. Basic demographic data, medical history, BMI, HRQOL, serum laboratory tests for metabolic health risk factors, including fasting glucose, triglyceride, and high-density lipoprotein cholesterol (HDL-C), and lifestyle habits, including smoking, alcohol consumption, and physical activity were evaluated at baseline. There were 427 participants who completed the follow-up evaluation between 2017 and 2018, including the repeated assessment of metabolic health, BMI, HRQOL, and other covariates. The mean follow-up period was 8.6 ± 0.6 years. However, participants with missing data at the follow-up assessment were still preserved in the analytical sample [17]. A total of 906 participants with complete data at baseline were included for the analyses.”

We also added the information of SF-36 at baseline in Table 1 (page 5-6) and the information of SF-36 at follow-up in the supplementary table (Table S1).

Reference 17: Ibrahim, J.G.; Molenberghs, G. Missing data methods in longitudinal studies: a review. Test (Madr). 2009, 18, 1-43.

2)    A Table with subject characteristics at follow-up would be helpful. 

Response: Thanks for the reviewer’s comment. We have added a supplementary table (Table S1) to describe the characteristics at follow-up of 427 participants with a complete follow-up assessment. We also added the information in the results (page 5, 1st paragraph) as follows:

“The demographic characteristics at follow-up of 427 participants with complete follow-up assessment are shown in Table S1”

3)    Comparing data to the MHNW group as the primary analysis makes sense but a secondary analysis comparing MHO and MUO would be helpful to determine whether outcomes differ between those groups. 

Response: Thanks for your suggestions. We conducted the analyses to compare the HRQOL between metabolically healthy overweight and metabolically unhealthy overweight, as well as metabolically healthy obesity and metabolically unhealthy obesity. The results were attached in the supplementary table (Table S2). Compared with individuals of metabolically healthy obesity, those of metabolically unhealthy obesity had lower scores of physical component summary, bodily pain, and general health. We also added the information on page 7, 1st paragraph as follows:

"The comparisons of HRQOL between individuals with metabolically healthy overweight and metabolically unhealthy overweight, as well as metabolically healthy obesity and metabolically unhealthy obesity are shown in Table S2."

4)    In methods age range is 35-55 years but 30 to 60 yr data is presented in Table 1.

Response: Thanks for the reviewer’s correction. We have corrected the range of age in Table 1 (page 5, Table 1) as 35-39, 40-49, and 50-55 years.

5)    In the introduction it is stated that “Metabolically healthy obesity (MHO) refers to obesity free from hypertension, hyperglycemia, and dyslipidemia.” As mentioned by the authors there is no universally used criteria for MHO. Please add “typically” before “refers to” or some similar wording and provide a reference at the end of this sentence.

Response: Thanks for the reviewer’s suggestion. We have revised this sentence and added a reference as follows (page 2, 1st paragraph):

“Metabolically healthy obesity (MHO) typically refers to obesity free from hypertension, hyperglycemia, and dyslipidemia [6].”

6)    In the introduction it is mentioned that “MHO does not mean truly healthy; rather, it refers to having fewer metabolic syndrome components than healthy people.” Again this statement requires some nuance with “does not always mean” instead of “does not mean” and “typically” before “refers to” or similar wording added.

Response: Thanks for the reviewer’s suggestion. We have revised this sentence and added a reference as follows (page 2, 2nd paragraph):

“However, MHO does not always mean truly healthy; rather, it typically refers to having fewer metabolic syndrome components than healthy people [7].”

Reviewer 2 Report

The authors conducted a prospective cohort study to assess the association between metabolic health/BMI category and health-related quality of life (HRQOL) in the general population aged 35-55 years living in Taiwan. In particular, the authors’ main focus was on HRQOL in individuals with metabolically healthy obesity MHO). The scores of physical component summary (PCS) and mental component summary (MCS) of the SF-36 were not significantly different between individuals with MHO and those with metabolically healthy normal weight (MHNW).

Major comments

  1. There are few reports on the association between MHO and HRQOL, and none have been reported in Asians. However, it is recommended that the authors provide more details on why it is important to consider HRQOL in MHO.

  1. The authors criticized the study by Lopez-Garcia et al. for not taking into account changes in metabolic health and body mass index. The authors should explain why those changes should be taken into account.

  1. The number of study participants in the study seems to be too small to compare the six different categories. At baseline, only 59 subjects were enrolled in the MHO; at the end of the 8-year follow-up, the number of subjects would have been even lower.

  1. The age of study participants ranged from 35 – 55 years. What was the reason for narrowing down the age range of the subjects?

  1. I am wondering why the authors chose an 8-year follow-up period for this study.

  1. How often were HRQOL, BMI, and metabolic status measured during the follow-up period? And how were these data during the follow-up period handled in the statistical analysis?

  1. The authors discuss the fact that the scores were better in MHO compared to MHNW for bodily pain, citing the article by Price RC, et al (Ref. 31). Usually, obese people are more likely to experience pain in the bones and joints, but the Price RC, et al. paper did not evaluate pain thresholds in the bone and joint region.

  1. The authors described the criteria of metabolic health: (A) absence of hypertension, hyperlipidemia, type 2 diabetes, coronary artery diseases, cerebral vascular diseases, and peripheral vascular diseases. The authors should describe the definition of hypertension, hyperlipidemia, and type 2 diabetes. In (c) and (d) in criteria (B), what drugs were used for the treatment of hypertriglyceridemia and low HDL-C? Although high LDL-C was not included in the criteria, were the participants who were prescribed statins classified as dyslipidemia?

  1. In the abstract and the conclusion, the authors state that this study showed that metabolically healthy individuals with obesity and normal weight had similar HRQOL. However, in each domain, there may be differences in HRQOL between MHO and MHNW.

Minor

  1. In table 1, age was categorized into 30-39, 40-49, and 50-59. However, the age of study participants ranged from 35 to 55 years. This age grouping may mislead readers.

Author Response

On behalf of all the authors, we thank you very much for reviewing our manuscript. During the major revision, we found some mistakes in SAS code in the main analyses for domains of SF-36. We have corrected the errors. Fortunately, the significance of the statistical analyses and results are the same and only the effect size is altered. We revised Figure 1 and Table 4 accordingly. Thank you again for your valuable comments, which really helped us to improve this article. Please find below our point-by-point responses to the comments from the reviewers. Revisions were made by using track changes in the revised manuscript.

Response to reviewers’ comments:

Reviewer #2:

Comments and Suggestions for Authors

The authors conducted a prospective cohort study to assess the association between metabolic health/BMI category and health-related quality of life (HRQOL) in the general population aged 35-55 years living in Taiwan. In particular, the authors’ main focus was on HRQOL in individuals with metabolically healthy obesity MHO). The scores of physical component summary (PCS) and mental component summary (MCS) of the SF-36 were not significantly different between individuals with MHO and those with metabolically healthy normal weight (MHNW).

Major comments

  1. There are few reports on the association between MHO and HRQOL, and none have been reported in Asians. However, it is recommended that the authors provide more details on why it is important to consider HRQOL in MHO.

Responses: Thanks for the reviewer’s comment. Health is a state of complete physical, mental and social well-being, which is a comprehensive concept that could be evaluated by HRQOL. HRQOL represents an overall assessment of physical, mental, and social aspects of one's well-being. Many studies about obesity reported different patterns of impaired physical and mental aspects on HRQOL among obese people. However, research about MHO and HRQOL is rare, which is the research gap we are interested in. We have added more information to explain why it is important to consider HRQOL in MHO (page 2, 3rd paragraph).

“Health is not merely free from physical illness, mental and social well-being are also essential components. Health-related quality of life (HRQOL) represents a comprehensive evaluation of physical, mental, and social aspects of one's wellbeing...... A past meta-analysis showed different patterns of impaired physical and mental aspects of HRQOL in people with obesity [13]. However, research on MHO and HRQOL is rare.”

  1. The authors criticized the study by Lopez-Garcia et al. for not taking into account changes in metabolic health and body mass index. The authors should explain why those changes should be taken into account.

Response: Many cohort studies about MHO found the status of MHO changed over time. The rate of transition from a metabolically healthy to unhealthy status for obese people ranged from 32.0% to 48 %, and it increased with the extension of the follow-up period [1-3]. The past study about MHO and cardiovascular disease showed poorer outcomes would be easily observed when MHO was only evaluated at baseline [4]. Other studies considered the change of metabolic health and BMI over time and showed insignificant associations between a sustained MHO status and incident diabetes and cardiovascular events [1, 2]. Therefore, the changes of MHO should be considered in cohort studies to avoid potential bias of misclassification for MHO

Reference:

    1. Appleton SL, et al. 2013. Diabetes and cardiovascular disease outcomes in the metabolically healthy obese phenotype: a cohort study. Diabetes Care 36(8):2388-2394.
    2. Gao M, et al. 2020. Metabolically healthy obesity, transition to unhealthy metabolic status, and vascular disease in Chinese adults: A cohort study. PLoS Medicine 17(10):e1003351.
    3. Mongraw-Chaffin M, et al. 2018. Metabolically healthy obesity, transition to metabolic syndrome, and cardiovascular risk. Journals of the American College of Cardiology 71(17):1857-1865.
    4. Yeh TL, et al. 2021. Association between metabolically healthy obesity/overweight and cardiovascular disease risk: A representative cohort study in Taiwan. PLoS One 16(2):e0246378.
  1. The number of study participants in the study seems to be too small to compare the six different categories. At baseline, only 59 subjects were enrolled in the MHO; at the end of the 8-year follow-up, the number of subjects would have been even lower.

Response: Thanks for the reviewer’s comment. The number of participants in the subgroup of MHO was 59 at baseline and decreased to 21 at follow-up (Table S1). We have added the limitation in the discussion (page 12, 2nd paragraph):

 “Third, the number of study participants in the subgroups of metabolic health/BMI was small. …… Further research with a larger representative cohort is needed.”

  1. The age of study participants ranged from 35 – 55 years. What was the reason for narrowing down the age range of the subjects?

Response: The study was part of a research project, "The Taipei Veteran General Hospital Cohort study of Cardiovascular Metabolic Risk Factors in Shihpai Area in Taiwan". The research project aimed to explore the change of metabolic disorders and the associated risk factors among community-dwelling adults. Due to the limited research funding, the research project enrolled the study participants aged 35 – 55 years. It is because lifestyle modification would be more effective in middle-aged people than older adults. We would like to broaden the age group of study participants if we could get more research resources in the future.

  1. I am wondering why the authors chose an 8-year follow-up period for this study.

Response: The study was part of a research project "The Taipei Veteran General Hospital Cohort study of Cardiovascular Metabolic Risk Factors in Shihpai Area in Taiwan". The research project aimed to observe the change of metabolic health, BMI, and health behaviors during a follow-up period of 8 years. For most cohort studies, the longer, the better. However, the attrition rates also increased with the extension of the follow-up period. More resources were needed to be devoted to increasing the retention rate in a cohort study with a longer follow-up period. Because of the limited research funding, we could only follow the participants during a follow-up period of 8 years. Further research with a longer follow-up duration would be valuable to investigate the long-term effects of MHO on HRQOL.

  1. How often were HRQOL, BMI, and metabolic status measured during the follow-up period? And how were these data during the follow-up period handled in the statistical analysis?

Response: There were 906 participants enrolled at baseline and they received the evaluation of metabolic health, BMI, HRQOL, and other covariates at baseline. After a mean follow-up period of 8.6 ± 0.6 years, 427 participants came back for follow-up. These 427 participants completed the follow-up assessment of metabolic health, BMI, HRQOL, and other covariates. However, participants with missing data at the follow-up assessment were still preserved in the analytical sample because we used the generalized linear mixed-effects models (GLMMs) for analyses. GLMMs contained fixed and random effects, which can control the correlation that arises from repeated measurement. GLMMs can also eliminate complete-case bias and allow for unbalanced data by incorporating all available information (Reference 17). We have added more description in the methodology (page 2, 4th paragraph) as follows:

“At baseline (2009-2010), 906 participants were enrolled in this study. Basic demographic data, medical history, BMI, HRQOL, serum laboratory tests for metabolic health risk factors, including fasting glucose, triglyceride, and high-density lipoprotein cholesterol (HDL-C), and lifestyle habits, including smoking, alcohol consumption, and physical activity were evaluated at baseline. There were 427 participants who completed the follow-up evaluation between 2017 and 2018, including the repeated assessment of metabolic health, BMI, HRQOL, and other covariates. The mean follow-up period was 8.6 ± 0.6 years. However, participants with missing data at the follow-up assessment were still preserved in the analytical sample [17]. A total of 906 participants with complete data at baseline were included for the analyses.”

Reference 17: Ibrahim, J.G.; Molenberghs, G. Missing data methods in longitudinal studies: a review. Test (Madr). 2009, 18, 1-43.

  1. The authors discuss the fact that the scores were better in MHO compared to MHNW for bodily pain, citing the article by Price RC, et al (Ref. 31). Usually, obese people are more likely to experience pain in the bones and joints, but the Price RC, et al. paper did not evaluate pain thresholds in the bone and joint region.

Response: Thanks for the reviewer’s comment. Price RC, et al reported the sensitivity of pain decreased on the abdomen but not altered on the forehead and hand in obese people. As your valuable comments, the locations and kinds of pain should be considered for obese people. We have added more information in this section (page 11, 5th paragraph) as follows:

“However, Price et al. also observed the sensitivity of pain on the forehead and hand was not altered in obese participants. In our study, the questionnaire of SF-36 evaluated the general perception of pain without the differentiation for the location of pain. Further studies are needed to investigate the perception of pain among people of MHO with considerations of the kinds and locations of pain.”

  1. The authors described the criteria of metabolic health: (A) absence of hypertension, hyperlipidemia, type 2 diabetes, coronary artery diseases, cerebral vascular diseases, and peripheral vascular diseases. The authors should describe the definition of hypertension, hyperlipidemia, and type 2 diabetes. In (c) and (d) in criteria (B), what drugs were used for the treatment of hypertriglyceridemia and low HDL-C? Although high LDL-C was not included in the criteria, were the participants who were prescribed statins classified as dyslipidemia?

Response: In this study, participants who were prescribed statins, as well as the combination of ezetimibe/statin and niacin/statin, were classified as the group of on drug treatment of low HDL-C. As the reviewer’s comment, statins may also be used to treat high LDL-C, or mixed dyslipidemia, which can lead to misclassification. However, there is no drug specific to treat low HDL-C. So the misclassification seems to be inevitable. In addition, patients with metabolic syndrome usually have increased small LDL-C particles and a reduced level of HDL-C [1]. Higher dense LDL-C was also highly correlated with an increased risk of cardiovascular diseases [2] and atherosclerosis, which implied the misclassification seems to be theoretically tolerable.

Reference:

  1. Grundy SM et al. Diagnosis and management of the metabolic syndrome: an American Heart Association/National Heart, Lung, and Blood Institute Scientific Statement. Circulation. 2005, 112, 2735-2752.
  2. Liou L, Kaptoge S. Association of small, dense LDL-cholesterol concentration and lipoprotein particle characteristics with coronary heart disease: A systematic review and meta-analysis. PLoS ONE 15(11):e0241993

  1. In the abstract and the conclusion, the authors state that this study showed that metabolically healthy individuals with obesity and normal weight had similar HRQOL. However, in each domain, there may be differences in HRQOL between MHO and MHNW.

Response: Thanks for the reviewer’s comment. In our results, there were some differences in domains of HRQOL between individuals with MHO and MHNW. The performance of PCS and MCS were similar between these two groups. Therefore, we concluded, “individuals with MHO had similar HRQOL in both PCS and MCS, compared with people with MHNW”. We modified the abstract as follows (page 1):

“This study showed that metabolically healthy individuals with obesity and normal weight had similar HRQOL in physical and mental component summary scores”.

Minor

  1. In table 1, age was categorized into 30-39, 40-49, and 50-59. However, the age of study participants ranged from 35 to 55 years. This age grouping may mislead readers.

Response: Thanks for the reviewer’s correction. We have corrected the range of age in Table 1 (page 5, Table 1) as 35-39, 40-49, and 50-55 years.

Round 2

Reviewer 1 Report

The authors have addressed most of my concern but I am still unclear why follow-up data was included? The only analysis I can see presented in the Results section is the prevalence of MHO at each time point. As the focus of the manuscript is health-related quality of life, information on whether this was stable over time and differed between groups at follow-up should be added to the Results section. 

Author Response

Submission ID: jcm-1387209

On behalf of all the authors, we thank you very much for reviewing our manuscript. Your valuable comments do help us to improve this article. Please find below our point-by-point responses to the comments from the reviewers. Revisions were made by using track changes in the revised manuscript.

Response to reviewers’ comments:

Reviewer #1:

Comments and Suggestions for Authors

  1. The authors have addressed most of my concern but I am still unclear why follow-up data was included? The only analysis I can see presented in the Results section is the prevalence of MHO at each time point. As the focus of the manuscript is health-related quality of life, information on whether this was stable over time and differed between groups at follow-up should be added to the Results section.

Response:

Thanks for your comments. We have added the description on the health-related quality of life in Results (page 5, 1st paragraph) as following:

“At baseline, participants with MHO had higher scores of PCS, role-physical, bodily pain, general health, and social functioning. The demographic characteristics at follow-up of 427 participants with complete follow-up assessment are shown in Table S1. After the nine-year follow-up, the scores of vitality, social functioning, mental health, and MCS improved (mean score at baseline and follow-up: 47.8 to 54.0, 31.6 to 50.3, 38.5 to 47.0, and 38.0 to 48.7, respectively; Table 1 and Table S1). There were no significant differences in PCS, MCS, and the eight domains of HRQOL among different metabolic health/BMI groups at the follow-up assessment.”

Reviewer 2 Report

Thank you for responding to my comments. I still have some questions about the responses. I think that some of those are due to my lack of understanding of the methodology, especially in statistical analysis.

Comment on responses to Comment 2 and 5

According to the references provided by the authors, the effect of metabolic health/BMI status on cardiovascular disease was examined both at baseline metabolic health/BMI and when the same metabolic health/BMI status persisted during the follow-up period. It is reasonable to prospectively evaluate metabolic health/BMI and risk of cardiovascular disease. However, the authors examined the association between metabolic health/BMI and HRQOL. Did the authors want to examine how the current metabolic health/BMI category would affect future health-related quality of life in the future, rather than the present? If so, why was future health-related QOL important? Was there any difference between metabolic health/BMI and health-related quality of life at baseline and at the end of follow-up? How did the authors classify the participants whose metabolic health/BMI status changed between baseline and the end of follow-up?

  1. Comment on response to Comment 8

As the authors’ response, statins and/or ezetimibe are usually used to treat hyper LDL-cholesterolemia and there are no drugs that treat only low HDL-C. Is there a diagnostic criteria for metabolic syndrome that classifies statins and/or ezetimibe use as low HDL-C? Or have previous studies on metabolically healthy obesity also classified statin and/or ezetimibe use as low HDL-C?

Author Response

Submission ID: jcm-1387209

On behalf of all the authors, we thank you very much for reviewing our manuscript. Your valuable comments do help us to improve this article. Please find below our point-by-point responses to the comments from the reviewers. Revisions were made by using track changes in the revised manuscript.

Response to reviewers’ comments: 

Reviewer #2:

Comments and Suggestions for Authors

Thank you for responding to my comments. I still have some questions about the responses. I think that some of those are due to my lack of understanding of the methodology, especially in statistical analysis.

  1. Comment on responses to Comment 2 and 5

According to the references provided by the authors, the effect of metabolic health/BMI status on cardiovascular disease was examined both at baseline metabolic health/BMI and when the same metabolic health/BMI status persisted during the follow-up period. It is reasonable to prospectively evaluate metabolic health/BMI and risk of cardiovascular disease. However, the authors examined the association between metabolic health/BMI and HRQOL. Did the authors want to examine how the current metabolic health/BMI category would affect future health-related quality of life in the future, rather than the present? If so, why was future health-related QOL important? Was there any difference between metabolic health/BMI and health-related quality of life at baseline and at the end of follow-up? How did the authors classify the participants whose metabolic health/BMI status changed between baseline and the end of follow-up?

Response: Thanks for your suggestions. We analyzed the association between baseline metabolic health/BMI and baseline HRQOL by multiple linear regression models (table below, model A). Compared with individuals of metabolically healthy normal weight, the positive scores of bodily pain among people with MHO in the main analysis became insignificant. We further conducted the linear regression models for the association between baseline metabolic health/BMI and follow-up HRQOL with adjustment for baseline HRQOL (table below, model B). The higher scores of bodily pain and general health among people with MHO became insignificant. These results were not completely identical to the main text because the metabolic health/BMI status and HRQOL of participants changed over time.

The prevalence of MHO in study participants was 6.5% at baseline and decreased to 4.9% at eight-year follow-up (page 5, 1st paragraph). The prevalence of metabolically unhealthy obesity was 10.6% at baseline and increased to 13.8% at follow-up (Table 1 and Table S1). At baseline, participants with MHO had higher scores of PCS, role-physical, bodily pain, general health, and social functioning (Table 1). At the follow-up assessment, there were no significant differences in PCS, MCS, and eight domains of HRQOL among different metabolic health/BMI groups (Table S1). Because the metabolic health/BMI status and HRQOL did change over time, we analyzed both the metabolic health/BMI status and HRQOL at baseline and follow-up assessment in the predictive models to make it closer to reality.

Model A. The association between the baseline metabolic health/BMI groups and baseline HRQOL by multiple linear regression models with adjustment for age, sex, marital status, level of education, smoking, alcohol consumption, and physical activity at baseline (n=906).

Metabolically healthy
normal weight

Metabolically healthy
overweight

Metabolically healthy
obesity

Metabolically unhealthy
normal weight

Metabolically unhealthy
overweight

Metabolically unhealthy
obesity

PCS

Ref.

-1.36 (-2.67, -0.05)

1.32 (-0.50, 3.13)

-3.22 (-4.61, -1.83)

-0.61 (-2.17, 0.94)

-1.95 (-3.51, -0.40)

MCS

Ref.

0.67 (-0.31, 1.65)

0.21 (-1.14, 1.57)

0.67 (-0.37, 1.71)

0.84 (-0.32, 2.00)

0.91 (-0.25, 2.07)

Physical functioning

Ref.

-1.53 (-2.48, -0.58)

-1.28 (-2.60, 0.04)

-1.88 (-2.89, -0.87)

-1.02 (-2.15, 0.11)

-1.29 (-2.41, -0.16)

Role-physical

Ref.

-1.68 (-3.33, -0.04)

1.86 (-0.41, 4.14)

-2.02 (-3.77, -0.28)

0.67 (-1.28, 2.62)

-0.85 (-2.79, 1.10)

Bodily pain

Ref.

-0.29 (-1.89, 1.32)

1.72 (-0.51, 3.94)

-2.93 (-4.64, -1.22)

0.53 (-1.38, 2.44)

-1.42 (-3.32, 0.49)

General health

Ref.

-0.23 (-2.01, 1.54)

3.43 (0.97, 5.88)

-3.34 (-5.23, -1.46)

-0.86 (-2.96, 1.25)

-2.20 (-4.30, -0.09)

Vitality

Ref.

0.07 (-0.83, 0.96)

-1.59 (-2.83, -0.35)

-0.98 (-1.93, -0.03)

-0.50 (-1.56, 0.57)

-0.86 (-1.92, 0.20)

Social functioning

Ref.

0.54 (-0.28, 1.37)

1.53 (0.39, 2.67)

0.21 (-0.66, 1.09)

0.18 (-0.80, 1.15)

1.32 (0.35, 2.30)

Role-emotional

Ref.

-0.02 (-1.93, 1.88)

2.20 (-0.44, 4.84)

0.46 (-1.57, 2.48)

1.56 (-0.70, 3.82)

0.96 (-1.30, 3.22)

Mental health

Ref.

-0.14 (-1.02, 0.74)

-1.28 (-2.50, -0.07)

-0.77 (-1.70, 0.16)

0.25 (-0.79, 1.29)

-0.48 (-1.53, 0.56)

Model B. The association between the baseline metabolic health/BMI groups and follow-up HRQOL by multiple linear regression models with adjustment for age, sex, marital status, level of education, smoking, alcohol consumption, physical activity, and scores of SF-36 at baseline (n=906).

Metabolically healthy
normal weight

Metabolically healthy
overweight

Metabolically healthy
obesity

Metabolically unhealthy
normal weight

Metabolically unhealthy
overweight

Metabolically unhealthy
obesity

PCS

Ref.

-0.25 (-1.98, 1.48)

-1.33 (-3.53, 0.87)

0.14 (-1.72, 1.99)

0.57 (-1.44, 2.58)

-2.26 (-4.27, -0.26)

MCS

Ref.

-0.66 (-3.12, 1.80)

1.97 (-1.17, 5.10)

-3.25 (-5.87, -0.63)

-2.34 (-5.21, 0.53)

-2.55 (-5.39, 0.30)

Physical functioning

Ref.

0.44 (-0.93, 1.81)

-1.28 (-3.00, 0.43)

0.59 (-0.88, 2.07)

0.05 (-1.54, 1.65)

-2.06 (-3.66, -0.46)

Role-physical

Ref.

-0.39 (-2.86, 2.07)

-0.22 (-3.37, 2.94)

-2.25 (-4.96, 0.46)

0.85 (-2.09, 3.78)

-3.52 (-6.44, -0.60)

Bodily pain

Ref.

-0.77 (-3.00, 1.45)

1.02 (-1.81, 3.85)

0.003 (-2.42, 2.43)

-0.17 (-2.80, 2.46)

-0.78 (-3.40, 1.83)

General health

Ref.

-1.35 (-3.36, 0.67)

-1.77 (-4.38, 0.85)

-1.48 (-3.69, 0.73)

-1.24 (-3.64, 1.16)

-3.89 (-6.27, -1.51)

Vitality

Ref.

0.37 (-1.82, 2.56)

1.95 (-0.91, 4.81)

-1.55 (-3.96, 0.86)

-1.94 (-4.53, 0.65)

-1.89 (-4.48, 0.70)

Social functioning

Ref.

-1.71 (-3.51, 0.09)

-0.84 (-3.18, 1.51)

-2.35 (-4.33, -0.37)

0.01 (-2.13, 2.16)

-2.83 (-4.97, -0.70)

Role-emotional

Ref.

-0.20 (-2.90, 2.51)

0.05 (-3.40, 3.51)

-3.21 (-6.16, -0.25)

-1.28 (-4.50, 1.94)

-4.46 (-7.65, -1.27)

Mental health

Ref.

-1.03 (-3.44, 1.38)

2.49 (-0.59, 5.58)

-2.47 (-5.12, 0.19)

-2.21 (-5.07, 0.66)

-0.82 (-3.68, 2.03)

  1. Comment on response to Comment 8

As the authors’ response, statins and/or ezetimibe are usually used to treat hyper LDL-cholesterolemia and there are no drugs that treat only low HDL-C. Is there a diagnostic criteria for metabolic syndrome that classifies statins and/or ezetimibe use as low HDL-C? Or have previous studies on metabolically healthy obesity also classified statin and/or ezetimibe use as low HDL-C?

Response: Thanks for your comments. According to the publication by the American Heart Association (Grundy 2005), therapeutic recommendations for low HDL-C in metabolic syndrome put priority on the LDL-C-lowering drug therapy (statins). In the past literature about MHO, lipid-lowering medication or cholesterol medication was regarded as a treatment for low HDL-C (Appleton 2013; Camhi 2013). Another review article on the management of low HDL-C (Khera 2013) reported that statins remain the first-line therapy among individuals with low HDL-C levels. Therefore, it is reasonable that statins were regarded as a drug treatment for low HDL-C in this study. In our raw data, one participant was prescribed the combination drug of statin/ezetimibe so this participant was also classified as receiving treatment for low HDL-C.

Reference:

Grundy, S.M. et al. Diagnosis and management of the metabolic syndrome: an American Heart Association/National Heart, Lung, and Blood Institute Scientific Statement. Circulation. 2005, 112, 2735-2752.

Appleton, S.L. et al. Diabetes and cardiovascular disease outcomes in the metabolically healthy obese phenotype: a cohort study. Diabetes Care. 2013, 36, 2388-2394.

Camhi, S.M. et al. Physical Activity and Screen Time in Metabolically Healthy Obese Phenotypes in Adolescents and Adults. Journal of Obesity. 2013:984613.

Khera, A.V. et al. Management of Low Levels of High-Density Lipoprotein-Cholesterol. Circulation. 2013;128:72–78.
